# Hyperendemicity of cysticercosis in Madagascar: Novel insights from school children population-based antigen prevalence study

Jean-François Carod[1]*, Frédéric Mauny[2,3], Anne Laure Parmentier[2,3], Maxime Desmarets[2], Mahenintsoa Rakotondrazaka[1], Alice Brembilla[3], Véronique Dermauw[4], Julien Razafimahefa[5], Rondro Mamitiana Ramahefarisoa[1], Marcellin Andriantseheno[5], Sarah Bailly[6], Didier Ménard[7], Pierre Dorny[4]

1 Laboratoire de Parasitologie, Pasteur Institute of Madagascar, Antananarivo, Madagascar, 2 Unité de méthodologie en recherche clinique, épidémiologie et santé publique, CIC Inserm 143, University Hospital Center, Besançon, France, 3 Laboratoire Chrono environnement, UMR 6249, CNRS-université de Franche Comté, Université de Bourgogne Franche-Comté, Besançon, France, 4 Department of Biomedical Sciences, Institute of Tropical Medicine, Antwerp, Belgium, 5 Neurology Unit, Hospital Befelatanana, Antananarivo, Madagascar, 6 Epidemiology Unit, Pasteur Institute of French Guiana, Cayenne, French Guiana, 7 Malaria Research Unit, Pasteur Institute of Madagascar, Antananarivo, Madagascar

* jfcarod@yahoo.es

**Data Availability Statement:** Data of the study belongs to the Pasteur Institute of Madagascar, BP 1274 Ambatofotsikely Avaradoha 101

## Abstract

### Objective

*Taenia solium* (Ts) cysticercosis is a neglected zoonotic disease particularly prevalent in Madagascar. Few data are available for children, current data mainly rely on antibody prevalence. We sought to determine the Ts-antigen seroprevalence–determining active cysticercosis—amongst school children from various cities in Madagascar (excluding the capital) and evaluated associated risk factors.

### Methods

In seven cities in Madagascar, the presence of cysticercosis in school children (n = 1751) was investigated in 2007 using the B158/B60 antigen (Ag)-ELISA.

### Results

The overall prevalence based on Ag detection was 27.7% [95%CI: 10–37%]. Risk factors associated with Ag positivity were age, biotope, altitude and annual average rainfall.

### Conclusion

These results highlight the high prevalence of active cysticercosis in Madagascar among school children in an urban setting. This high prevalence as well as the risk factors unraveled point to the emergency to implement appropriate Public Health measure son a national scale.

Antananarivo, Madagascar. Access to data is restricted for legal reasons. The data may be made available after obtaining approval from the National Ethical Committee of Madagascar. Request should be sent to: Comité Malgache d'Ethique pour les Sciences et les Technologies, Boîte Postale 6217, Rue Fernand Kasanga, Tsimbazaza, Antananarivo, 101, Madagascar. Phone: +261 20 2221084. The authors had no special access privileges others would not have.

**Funding:** Pasteur Institute of Madagascar provided the human ressources, the laboratory equipments and provided the lab disposables. Reagents were provided by ITM, Antwerp. No specific fundings were received for this study.

**Competing interests:** The authors have declared that no competing interests exist.

## Introduction

Cysticercosis is a neglected parasitic zoonosis caused by the larval stage (cysticercus) of the pork tapeworm *Taenia solium*. Development of these larvae in the human central nervous system can cause seizures and other neurological symptoms (neurocysticercosis, NCC). NCC has been described as the most frequently reported helminthic infection of the central nervous system and it is a major cause of acquired epilepsy in cysticercosis endemic regions; it is associated with considerable morbidity and 30% of seizure disorders are attributable to NCC [1, 2]. Human cysticercosis mainly affects poor communities, where poor hygiene and sanitation, free-ranging pigs are present and adequate meat inspection is lacking, such as areas of South America, Brazil, Central America and Mexico; China, the Indian subcontinent and South-East Asia; and sub-Saharan Africa [3–5]. In Africa, the prevalence of human cysticercosis ranges from 7.4% in South Africa to 20.5% in Mozambique [6–8] (both based on specific antibody (Ab) detection) and 21.6% in the Democratic Republic of Congo [9] (based on circulating Ag detection). A systematic review estimated that the prevalence of circulating *T. solium* antigens for Sub-saharan Africa was 7.30% (95% CI [4.23–12.31]) vs. 17.37% (95% CI [3.33–56.20]) for the antibody seroprevalence [10].

In Madagascar, situated 400 km from the coast of the African continent, studies undertaken between 1994 and 1999, indicated high exposure to *T. solium* [11–13], based on results of Ab-detecting ELISA [14], with confirmation by means of an Electro Immunotransfer Blot (EITB) [14–16]. Furthermore, a significant variation in seroprevalence was found between the different provinces: < 10% in coastal regions (Mahajanga and Toamasina), while around 20% in central highland regions (Ihosy, Ambositra and Mahasolo) [11–13]. No differences according to age groups or urban vs. rural residence were found, yet the seroprevalence was higher in women vs. men. In another study in Mahajanga, 12.5% of children aged between 2 and 4 years had antibodies to *T. solium* while this was 21.8% in children aged between 5 to 14 years (similar figures in adults) [12].

Neurocysticercosis can be considered the main cause of secondary childhood epilepsy in Madagascar being one of the most important foci in the world [17, 18]. Pediatricians of the Cenhosoa Military hospital in Antananarivo noted that epilepsy was present in over 80% of the cases of cysticercosis cases admitted in pediatrics [19]. Because of complications to control seizures and increased intracranial pressure, these children may have a less favorable prognosis [20]. All seroepidemiological studies on cysticercosis in Madagascar used antibody detecting techniques, which rather point to exposure to the parasite then to active cysticercosis [21]. Studies on the presence of active cysticercosis by measuring circulating parasite antigens, especially in children, are lacking. The aims of the current study therefore, were to investigate the prevalence of active cysticercosis in school children in seven cities in the country and study associated risk factors.

## Materials and methods

### Study design

Between February and March 2007, a population-based study was carried out in children from 3 to 16 years old attending school in seven major cities in Madagascar (Fig 1), within the context a large scale study aiming to unravel biological factors associated with the protection against *Plasmodium* infection [22]. The four biotopes of Madagascar are characterized by specific environmental factors such as, rainfall, temperature and altitude. These areas are inhabited by different ethnic groups [20] with similar or different origins and different cultural habits. Per biotope, two sites were selected (except for the highlands): Andapa and Farafangana in the Equatorial biotope: Miandrivazo and Maevatanana in the Tropical biotope;

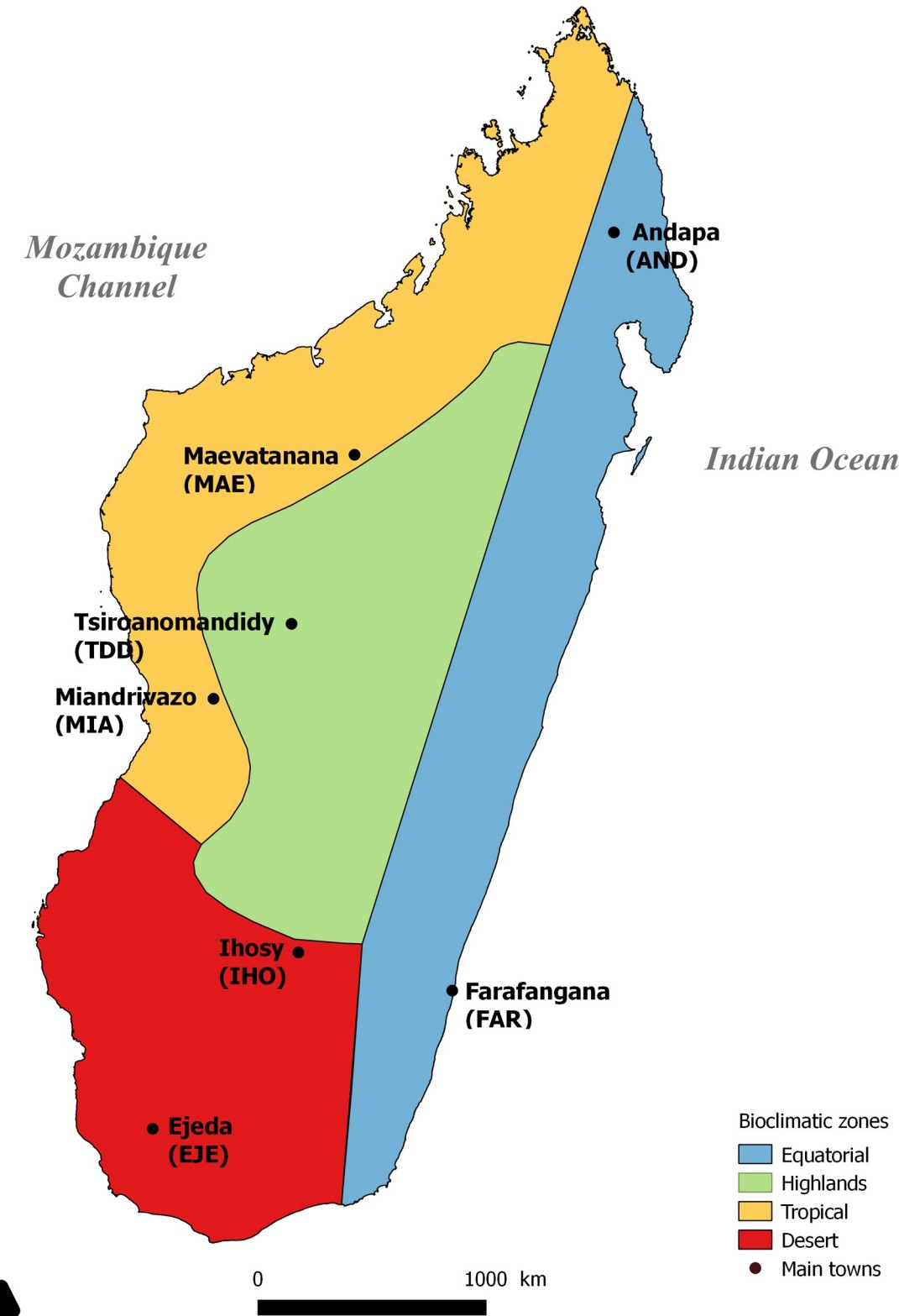

**Fig 1. Map of Madagascar indicating bioclimatic zones and major cities including the study sites.**

Tsiroanomandidy in the Highlands biotope; and Ihosy and Ejeda in the Desert biotope. In each town, 250 children were selected using a two-level cluster random sampling (school and classroom). School children from either public or private schools were selected. Depending on the site (= town), samples were collected from one (in Tsiroanomandidy and Miandrivazo), two (Andapa, Ejeda) or three schools (Farafangana, Ihosy and Maevatanana).

## Data and specimen collection

After obtaining informed consent from parents or guardians, the child was interviewed by a trained field worker. Several socio-demographic variables were recorded, such as, age, sex, place of residence, and the community and ethnic group of the child and his/her parents. The different communities were malagasy, comorian, caucasian, indo-pakistanese or Chinese. Malagasy children were divided into 20 ethnic groups. After the interview, blood (5 ml) was collected by venipuncture into an EDTA coated tube as well as in an additive-free tube. Samples were transported at +4˚C in a cool-box to the Pasteur Institute in Antananarivo (IPM), Madagascar where they were kept frozen at -20˚C until analysis. The analysis were performed in 2008. All samples were de-identified.

## Analytical procedures

The presence of circulating cysticercus antigens was measured by the monoclonal antibody-based B158/B60 Ag-ELISA [21, 23]. Sera from two known highly positive patients were used as positive controls. The optical density of each serum sample was compared to a set of 8 negative sera at a probability level of $p = 0.001$ to determine the result of the test. The test has a sensitivity of 90% (95% CI: 80%–99%) and a specificity of 98% (95% CI: 97%–99%), based on Bayesian analyses used by Praet et al. [10]. All reagents were produced at the Institute of Tropical Medicine (ITM), Antwerp including positive and negative control. The assays were performed at IPM, Antananarivo and all results were validated by ITM, Antwerp.

## Statistical analyses

To take into account the hierarchical structure of the data, the relationship between the individual serological status and the individual and collective factors was analyzed using a multi-level logistic regression model [24]. In this model, the child was level 1, while the city was level 2. Each factor was first analyzed in a univariable model. When significant at $p < 0.20$, factors were introduced in multivariable analysis using a backward stepwise procedure. The level of significance was set to 0.05. Since biotopes are highly associated with the climate patterns, two models have been used for the multivariable analysis. Sex, age, biotope, altitude, temperature and rainfall were introduced in the models as fixed effects. The model 1 integrated the different biotopes but didn't include rainfall, temperature or altitude; model 2 included the three previous patterns but excluded the different biotopes. All collected data was entered in an Excel (Microsoft Office Excel 2007Ⓡ) spreadsheet. Classical statistical analyses were conducted using R version 3.5.0. Multilevel analyses were performed using MLwiN V2.20 [25]. Interaction between variables in models were tested. None of them were significant or improved the models.

## Ethics

The current study has been reviewed and approved by the National Ethical Committee of Madagascar. Informed and written consent was obtained from the parents of children. The sera are no longer linked to personal identifiers.

**Table 1. Geo-climatic data and *Taenia solium* antigen prevalence assessed by B158/B60 Ag-ELISA of 7 sampling sites in Madagascar.**

|  | Highlands | Tropical |  | Equatorial |  | Desert |  |
|---|---|---|---|---|---|---|---|
|  | TDD (n = 297) | MAE (n = 250) | MIA (n = 295) | FAR (n = 267) | AND (n = 225) | EJE (n = 123) | IHO (n = 294) |
| Alt.(m) | 900 | 79 | 62 | 6 | 474 | 275 | 567 |
| Pluv.(mm) | 1618 | 1980 | 1363 | 2421 | 1995 | 567 | 730 |
| Temp (˚C) | 22.6 | 25.8 | 26.7 | 23.5 | 21.9 | 24.5 | 21.9 |
| % Ag+[a] (95% CI) | 36.7(31.2,42.5) | 28.8(23.3,34.9) | 30.2(25.0,35.8) | 20.6(16.0,26.0) | 10.2(6.7,15.1) | 31.7(23.7,40.7) | 33.7(28.3,39.4) |

[a] Missing value by sites: TDD = 9 ; MAE = 53 ; MIA = 1 ; FAR = 8 ; AND = 63 ; EJE = 24 ; IHO = 3.

AND:Andapa,EJE:Ejeda,FAR:Farafangana,IHO:Ihosy,MAE:Maevatanana,MIA:Miandrivazo, TDD: Tsiroanomandidy, Alt(m): altitude, Pluv: annual rainfall, Temp(˚C): average annual temperature, %Ag+: *Taenia solium* Antigen prevalence.

Sources: Direction générale de la météorologie à Madagascar for the Altitude criteria and Climate-data.org for the annual rainfall or temperature.

## Results

A total of 1751 schoolchildren were included in this study. The age ranged from 3to16 years, the M/F sex ratio was 1.02;53%(n = 924 children). Twenty ethnic groups are represented in the study population. Most of the children were of Merina or Betsileo ethnic group."

The average Ts Antigen prevalence was 27.7%, and ranged from 10 to37%. The equatorial sites had the lowest prevalence ranges (10–21%) while rates above 30% were found in the southern desert as well as in the highlands (Table 1).

The relation between age, sex and *T.solium* antigen seropositivity status was presented in the Table 2. Age was collected as a continuous variable, and was then treated as a dichotomous variable. We used the population mean as a cut-off. Only the age variable has a significant relationship with seropositivity status. However, as major epidemiological factors, age and sex were introduced in the univariable and multivariable analysis (Table 3).

### Univariable and multivariable multilevel logistic regression model analysis

Model 1 indicated a significant association between Ts Ag positivity and biotope (p<10−3), the Highlands biotope was associated with the highest prevalence. In Model 2, a significant association was found between Ts Ag positivity and both average annual temperature (p = 0.01) and altitude (p = 0.01) (Table 3).

## Discussion

The seroprevalence of cysticercosis based on antibody detection ranked Madagascar within the highly endemic countries. The lack of sensitivity of Ab-ELISA's may have underestimated the prevalence rates published in Madagascar. Most NCC cases (the most common clinical form of cysticercosis) found on the island are related to single cyst carriers, which are known to be difficult to detect with both antibody and antigen-based serological methods [26, 27].

**Table 2. Age and sex as possible risk factors for *T. solium* antigen seropositivity.**

| Characteristic | Categories | Total | | Ag + | | Ag - | | |
|---|---|---|---|---|---|---|---|---|
|  |  | n | % | n | % | n | % | P-Value |
| **Sex** | Male | 887 | 50.83 | 250 | 51.76 | 637 | 50.48 | 0.63 |
|  | Female | 858 | 49.17 | 233 | 48.24 | 625 | 49.52 |  |
| **Age (years)** | < 8 | 799 | 45.79 | 191 | 39.54 | 608 | 48.18 | <0.001 |
|  | ≥ 8 | 946 | 54.21 | 292 | 60.46 | 654 | 51.82 |  |

**Table 3. Univariable and multivariate analysis related to *T. solium* antigen positivity.**

| | Univariable analysis | | | Final multivariable analysis | | | | | | | |
| | | | | Model 1 | | | | Model 2 | | | |
| | OR | 95% CI | | p-value | OR | 95% CI | | p-value | OR | 95% CI | | p-value |
|---|---|---|---|---|---|---|---|---|---|---|---|---|
| Sex[a] (Ref = female) | 1.04 | 0.84 | 1.28 | 0.72 | 1.03 | 0.84 | 1.28 | 0.75 | 1.03 | 0.83 | 1.27 | 0.8 |
| Age[a] (Ref = <8 years old) | 1.28 | 1.02 | 1.60 | 0.03 | 1.24 | 0.99 | 1.55 | 0.06 | 1.26 | 1.00 | 1.57 | 0.05 |
| Biotope | | | | | | | | | | | | |
| Equatorial (Ref.) | 1.00 | | | <10−3 | 1.00 | | | <10−3 | / | / | / | / |
| Highlands | 3.08 | 2.19 | 4.31 | | 2.84 | 1.95 | 4.16 | | / | / | / | / |
| Tropical | 2.23 | 1.64 | 3.02 | | 2.13 | 1.53 | 2.97 | | / | / | / | / |
| Desert | 2.63 | 1.91 | 3.60 | | 2.54 | 1.80 | 3.58 | | / | / | / | / |
| Average annual rainfall [b] | 0.96 | 0.92 | 1.00 | 0.06 | / | / | / | / | | | | NS |
| Average annual temperature [c] | 1.06 | 0.88 | 1.27 | 0.53 | / | / | / | / | 1.28 | 1.06 | 1.54 | 0.01 |
| Altitude [d] | 1.04 | 0.95 | 1.14 | 0.37 | / | / | / | / | 1.14 | 1.03 | 1.26 | 0.01 |

[a]Missing data: sex (n = 6), age (n = 6)

b For 100 mm increase

c for 1˚C increase

d for 100 meters increase.

In contrast to the presence of Ts Ab, indicating exposure to the infection, the Ts Ag rate indicates the presence of active cysticercosis. With an average of 27% positives in children in our study, the Ts Ag prevalence rate exceeds all antigen prevalence rates described worldwide. High Ts Ag rates indicates that active infections with *T. solium* metacestodes occur within the Malagasy population, affecting the younger age class and that it is widespread on the island even though a visible heterogeneity was observed between the sites. Recent studies have shown that in African countries higher apparent prevalence of active infections occurred than in Asian countries [28]. This difference may reflect epidemiological differences, especially a higher exposure; however, variations in host susceptibility and in parasite genotypes may also play a role [29, 30].

Previous evidence based on antibody detection has indicated that all provinces of Madagascar were endemic for cysticercosis with significant differences between the areas. The use of the Ts Ag ELISA showed the same trends with a lower Ts Ag prevalence in the equatorial biotope. This supports the hypothesis that spatial heterogeneity in the distribution of infections may be influenced by environmental conditions, highlighting the interplay between socio-economic, behavioural and environmental factors in *Taenia* [31]. Selection bias may also be taken into account. One major difference of our study is that most other studies were conducted in adults in which the effect of gender is often modified by age and is an indirect measure of behaviour. Biotopes are represented by solely one or two sites with sometimes marked variations in environmental or cultural behaviours. The access to fresh water is enhanced in the equatorial area with better hygiene practices, and less common pig raising and pork consumption than in the highlands [32]. These conditions and practices may play a role in limiting the disease's prevalence. While arid areas should conversely experience a lower survival of parasitic eggs [33], in the present study these areas were associated with high Ts Ag prevalence rates, just below the highlands' rate. When considering the effect of desiccation on *Taenia* egg survival [33], this high Ag prevalence was not expected and requires further study.

An effect of gender on prevalence was not observed [20], unlike in studies in Latin America and other African countries. This might be due to cultural differences between populations

[28]. However, age was reported as a significant risk factor in the bivariate model for the presence of circulating antigens as quoted in different reports [28, 34, 35]. Worldwide, one third of the total epilepsy cases arise in childhood and neurocysticercosis is a major cause, particularly in developing countries [35, 36]. However, presence of antigen doesn't necessarily signify the presence of a viable, well-established cysticercus infection. It could be indicative for short term partial establishment, and perhaps a 'transient' antigen presence should be considered [37]. A Zambian study demonstrated the rather short-term presence of circulating antigen in 31 participants. Whether this observation is due to a partial establishment of infection (immature cysticerci), or the establishment of cysticerci, followed by a quick degeneration (self cure?) could not be confirmed. The Ts Ag results in that study were provided as qualitative data. It may be that antigen levels (ratio) can add complementary information. Indeed, Mwape et al [37] observed that individuals who became seronegative were those that displayed low levels of circulating antigen. Therefore, serological results from field studies should be looked at critically. Individuals with positive test results shouldn't be by definition considered as infected with *T. solium*, but the possibility of a transient antigen response has to be taken into account.

The benefit of treating children positive for Ts antigen is still debated, and the medical interventions should be carried out in neurology Units. However, in Madagascar neurologists are available only in Antananarivo and Mahajunga, and not in the cities under study. Treating a positive case without a proper medical examination may impair and even endanger the child's life and by destroying viable cysts, it may enhance the brain seizures with a high risk of morbidity and even mortality [9].

## Limitations

This is a cross-sectional study in schoolchildren from seven sites in the four different biotopes of Madagascar. The samples were collected 13 years ago. It may be suspected that risk factors have changed. However, according to Word Bank data (accessed 12/21/2020): GDP per capita (current USD) decreased from 536 to 523 USD. Assuming that the risk factors are associated with poverty and haven't significantly changed, the obtained data may be extrapolated to the current years.

The main limitations of our study are the lack of clinical data and risk factors associated with cysticercosis.

- The blood samples were collected in the framework of a malaria study. Therefore, factors recorded were not specific to cysticercosis. Further and updated studies should rely on prospective studies focusing on social, behavioural, sanitary, economic determinants and the Taeniasis/Cysticercosis risk factors that may play a role in the spread of the disease and explain the differences between sites. Implementing specifically cysticercosis driven prospective studies will enable to show the dynamic nature of *T. solium* infections in the different biotopes and adapt accordingly human and animal health policies.

- Clinical features have to be taken into account and a possible follow-up should be offered to patients featuring a probable/possible diagnosis of NCC [38]. The presence of a positive Ag Ts is not sufficient to diagnose NCC since its positivity does not indicate a neurological localization and an important number of infections probably never fully establish, leading to possibly 'transient' antigen presence [37]. Furthermore, in humans, it is described that cysticerci may stay viable during years according to their localization [3]. Many factors, among which, the size of the (re) infection, the immune status of the host, age and sex play a determining role in the (non-) establishment of infection [39].
  Clinical features including epilepsy are critical to define a diagnosis of NCC added to

imaging data, laboratory findings and epidemiology patterns [38]. Many reports described cysticercosis-associated epilepsy in Madagascar [40]: pediatricians of the military hospital of Antananarivo reported that epilepsy was present in over 80% of the cases of NCC. An interesting finding is that the seroprevalence (Ts Ab) found among suspected cases was between 25 and 48% (ELISA+/-EITB) [40]. However, those studies were completed between 1991 to 1993 and did not use Ts Ag as the diagnostic tool. Therefore, unpublished data communicated in 2012, displayed a Ts Ag positivity of 73% among patients having clinical and imaging features attributable to NCC [41]. Clinical symptoms and biological results are, however, not sufficient for a diagnosis of NCC [38]. CT-scan/MRI are essential for this purpose: they are the sole way to assess the number, the neurological localisation and the status (viable/degenerated/calcified) of cysticerci. Unfortunately, these imaging tools are obviously not applicable and available for field studies.

## Conclusions

Findings of the present study indicate that human *T. solium* infection is hyper-endemic in all biotopes of Madagascar. Active cysticercosis was found in almost 30% of all surveyed school-children. This proportion is the highest figure reported to date followed by Kanobana report in 2011 [9]. Despite the age effect, which disappeared in the multivariable analysis, the data shown that *cysticercosis* is present with alive cysts (positive antigen level) in a considerable amount of the population in Madagascar including children with possible damaging consequences for their growth and schooling. Further understanding of the disease distribution and transmission burden in Madagascar is needed as well as strengthening capacity to address it. Correspondingly, cross-sectoral (stakeholders) control efforts at local, regional, national should be initiated using appropriate, sustainable approaches to decrease or eliminate the burden of the disease.

## Acknowledgments

The authors are grateful to all participants who volunteered to take part in this study. The cysticercosis working group in Madagascar is thanked for their technical support. We acknowledge the valuable contribution and expertise of Pr Romy Razakandrainibe (Rouen University, France) and Dr Claude Flamand (Pasteur Institute of French Guiana).

## Author Contributions

**Conceptualization:** Jean-François Carod, Anne Laure Parmentier, Didier Ménard.

**Data curation:** Jean-François Carod, Frédéric Mauny, Anne Laure Parmentier, Maxime Desmarets, Pierre Dorny.

**Formal analysis:** Jean-François Carod, Frédéric Mauny, Anne Laure Parmentier, Maxime Desmarets, Alice Brembilla, Véronique Dermauw, Julien Razafimahefa, Pierre Dorny.

**Funding acquisition:** Jean-François Carod.

**Investigation:** Jean-François Carod, Anne Laure Parmentier, Mahenintsoa Rakotondrazaka, Alice Brembilla, Véronique Dermauw, Rondro Mamitiana Ramahefarisoa, Marcellin Andriantseheno, Didier Ménard, Pierre Dorny.

**Methodology:** Jean-François Carod, Frédéric Mauny, Anne Laure Parmentier, Maxime Desmarets, Mahenintsoa Rakotondrazaka, Julien Razafimahefa, Rondro Mamitiana Ramahefarisoa, Marcellin Andriantseheno, Didier Ménard, Pierre Dorny.

**Project administration:** Jean-François Carod.

**Resources:** Jean-François Carod, Julien Razafimahefa, Rondro Mamitiana Ramahefarisoa, Marcellin Andriantseheno, Sarah Bailly, Didier Ménard, Pierre Dorny.

**Software:** Frédéric Mauny, Alice Brembilla, Sarah Bailly.

**Supervision:** Jean-François Carod, Anne Laure Parmentier, Pierre Dorny.

**Validation:** Jean-François Carod, Anne Laure Parmentier, Pierre Dorny.

**Visualization:** Jean-François Carod.

**Writing – original draft:** Jean-François Carod, Frédéric Mauny, Anne Laure Parmentier, Julien Razafimahefa, Pierre Dorny.

**Writing – review & editing:** Anne Laure Parmentier, Maxime Desmarets, Véronique Dermauw, Sarah Bailly, Didier Ménard, Pierre Dorny.

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
