## [Decision Letter · Decision Letter 0]

8 Dec 2020

PONE-D-20-27048

Hyperendemicity of Cysticercosis in Madagascar: Novel insights from school children population-based seroprevalence study

PLOS ONE

Dear Dr. Carod,

Thank you for submitting your manuscript to PLOS ONE. After careful consideration, we feel that it has merit but does not fully meet PLOS ONE’s publication criteria as it currently stands. Therefore, we invite you to submit a revised version of the manuscript that addresses the points raised during the review process.

The high prevalence of cysticercosis in school children in Madagascar is an interesting finding but must be properly validated. The reviewers recommend reconsideration of the manuscript after major revision and I invite you to address all the recommendations below. I concur with the reviewers that the authors did not present a manuscript that is technically sound and that the data presented do not support the conclusions reached. The authors must clarify how the test was validated and provide a detailed description of the appropriate controls used. It is very important to explain how the samples were manipulated and stored and specify where were the tests performed. Moreover, details on the timelines and sampling are needed. I recommend the authors to reassess the statistical analysis and properly backup their conclusions as recommended by the reviewers. Please discuss your findings in relation to known risk factors, such as tapeworm carriers, which have been shown to be crucial in the transmission of cysticercosis.

We look forward to receiving your revised manuscript.

Kind regards,

Fela Mendlovic, MSc, PhD

Academic Editor

PLOS ONE

Journal Requirements:

2. Thank you for stating the following in the Footnotes Section of your manuscript:

"This work was funded by the Pasteur Institute of Madagascar with the contribution of

ITM. The funders had no role in the study design, data collection and analysis, decision

to publish, or preparation of the manuscript."

"The author(s) received no specific funding for this work"

Reviewers' comments:

Reviewer's Responses to Questions

**Comments to the Author**

1. Is the manuscript technically sound, and do the data support the conclusions?

Reviewer #1: No

Reviewer #2: No

Reviewer #3: Partly

2. Has the statistical analysis been performed appropriately and rigorously? 

Reviewer #1: N/A

Reviewer #2: No

Reviewer #3: No

3. Have the authors made all data underlying the findings in their manuscript fully available?

Reviewer #1: No

Reviewer #2: Yes

Reviewer #3: Yes

4. Is the manuscript presented in an intelligible fashion and written in standard English?

Reviewer #1: Yes

Reviewer #2: Yes

Reviewer #3: No

5. Review Comments to the Author

Reviewer #1: This manuscript describes a serological survey of school children in 7 major cities of Madagascar for the presence of circulating antigen being indicative of infection with viable cysticerci of Taenia solium. They found that up to 37% of the children were positive, with an average prevalence of 27%. The authors point out that this is the highest prevalence described for anywhere in the world, after one village in DRC where a seroprevalence of 21% was recorded.

The data in the paper give no confidence that the serological results are valid. The claimed level of active infection is either a disaster of extraordinary proportions for the school children of Madagascar, or an indication that the serology was faulty. We are provided with no data whatsoever about how or where the serology was performed. Two references are given as the serological methodology, published by the last author. Was the serology performed in in Madagascar or in Belgium in the senior author’s laboratory? How can we know the performance characteristics of the assay outside the senior author’s laboratory if the tests were undertaken in Madagascar? What control samples were tested from known cases of cysticercosis and, more importantly, from persons who have no history of cysticercosis? Did the authors source any negative controls (at least persons with no history of symptoms suggestive of being infected with T. solium) from the Malagasy population, such as members of the project team?

In the complete absence of any information to support the claimed extraordinarily high prevalence of cysticercosis in Malagasy children, and the complete absence of any information about the negative human controls used when the tests were performed, or where the tests were undertaken, the paper conclusions cannot be regarded as robust.

Reviewer #2: The manuscript needs a MAJOR revision.

Major comments:

1. Abstract: Should make clear that the samples were from 13 years ago (assuming that is the case), so the conclusion might not be valid, unless they speculate that risk factors haven’t changed.

2. Study design is not clear, it is only a geographical description.

3. Timelines are not clear through the manuscript. When were the samples collected? When were the tests performed?

4. Analytical procedures:

4a. Which methods were used to develop the ELISA?

4b. Where were the reagents produced?

4c. Source of positive controls?

4d. Source of negative sera? Are they from the same population?

4e. Any positives confirmed as positives? (if tests were done after 13 years, I understand it might not have been possible, but timelines are not clear).

5. Statistical analyses:

5a. The analysis by age is not clear. Which groups were analysed? What are the results by group?

5b. Why try to relate prevalence to altitude and rainfall, and not to the well-known and necessary risk factors such as presence of backyard pigs and open defecation?

6. Results: Results by age are not clear.

7. Conclusion:

7a. If the serum is indeed from 2007, 13 years later, many epileptic cases would have been detected. Is this the case? What is the rate of epilepsy?

8. Ethics: need to justify why the children were not followed-up as NCC is potentially very dangerous (I assume it is because of timings, but as they are not specified, it is not clear)

Minor comments:

9. Line 54: Use original reference for 30% seizure disorders (Ndimubanzi et al, 2010)

10. Line 56: Andean area of South America is not correct (Argentina is not endemic for example). Better change to areas of South America.

11. Figure 1: There is no legend to interpret the colours. The dark blue is difficult to read. What does it mean 237 in the title?

12. Table 1: Only last row is Ag prevalence. Title is not appropriate. Delete % sign from first results column.

13. Editing – in several places spaces before brackets are missing.

Reviewer #3: This is a very interesting manuscript showing that cysticercosis is infecting children at a very high prevalence in urban areas of Madagascar. The authors should however consider the following to improve it

1) the authors should follow the STROBE guidelines for reporting observational studies in epidemiology.

2) the sampling frame and study sites are not clearly described; there is a mismatch between the text and Fig1, and Fig1 title is unclear. How many sites in the highlands? Why more than 2 sites in the other areas?

3) given the high prevalence of infeciton, a log-binomial model would be more appropriate here and make interpretation of ratios more straightforward. It is also unclear if the biotype was included as a level in the hierarchical model or as a variable (only 2 levels are mentioned in the description). Each level used in the cluster sampling should be included in the multi-level model. it is unclear if this was done or not.

4) More details on sampling of households and children within each town (is town=site?) are needed. At this point, it is unclear if towns were randomly selected and how children within town were selected. This could introduce selection bias.

5) It seems very odd for a child aged 3 years old to be considered as school aged. Do 3-year olds go to school in Madagascar?

6) Denominators and 95%CI should be reported in Table 1 for ease of reading and interpretation

7) Why was the age dichotomized and was effect modification evaluated? Were there any other confounders assessed? SES for one could be an important confounder of the impact of even the environmental factors. General sanitation even at the site level is also important to adjust for, it may be confounding the association with environmental factors. What about ethnicity? There needs to be a Table of univariate analyses. Following the STROBE guidelines would help in this regard.

8) a more solid discussion of the impact of the environmental factors is needed with strenghts and limitations

9) There needs to a paragraph on limitations

Specific comments

Line 60-61: please provide a range of prevalence found with AgELISA in SSA (like what is done for Ab). There is also a relatively recent systematic review of prev of HCC in SSA which could be used here (Coral Almeida 2015),

Could there be more than 1 child per household? If so, was this considered in the analyses?

Study design: the DATES of the survey most be reported

Table 2 and related analyses: re-run the analyses with the Equatorial region as a reference for easier interpretation.

Lines 175-176: this is not quite correct. The EITB performs a lot better in this situation, though not perfect. Also, the AgELISA also performs poorly when there is only one NCC cyst, so this argument is incorrect the way it is stated here.

Lines 206-209: the big difference here is that most other studies were conducted in adults where the effect gender is often modified by age and is an indirect measure of behavior. This should be noted here.

6. PLOS authors have the option to publish the peer review history of their article (what does this mean?). If published, this will include your full peer review and any attached files.

Reviewer #1: No

Reviewer #2: No

Reviewer #3: No

---

## [Author Response · Author response to Decision Letter 0]

13 Apr 2021

PLOS ONE

Hyperendemicity of Cysticercosis in Madagascar: Novel insights from a school population-based antigen prevalence study

--Manuscript Draft--

Manuscript Number: PONE-D-20-27048

Article Type: Research Article

Answers to the Reviewers.

Reviewer #1: This manuscript describes a serological survey of school children in 7 major cities of Madagascar for the presence of circulating antigen being indicative of infection with viable cysticerci of Taenia solium. They found that up to 37% of the children were positive, with an average prevalence of 27%. The authors point out that this is the highest prevalence described for anywhere in the world, after one village in DRC where a seroprevalence of 21% was recorded.

The data in the paper give no confidence that the serological results are valid. The claimed level of active infection is either a disaster of extraordinary proportions for the school children of Madagascar, or an indication that the serology was faulty. We are provided with no data whatsoever about how or where the serology was performed. Two references are given as the serological methodology, published by the last author. Was the serology performed in in Madagascar or in Belgium in the senior author’s laboratory? How can we know the performance characteristics of the assay outside the senior author’s laboratory if the tests were undertaken in Madagascar? What control samples were tested from known cases of cysticercosis and, more importantly, from persons who have no history of cysticercosis? Did the authors source any negative controls (at least persons with no history of symptoms suggestive of being infected with T. solium) from the Malagasy population, such as members of the project team?

In the complete absence of any information to support the claimed extraordinarily high prevalence of cysticercosis in Malagasy children, and the complete absence of any information about the negative human controls used when the tests were performed, or where the tests were undertaken, the paper conclusions cannot be regarded as robust.

We acknowledge that the Ag-ELISA used in the current study is an in-house test and therefore, we understand the concerns raised by reviewer #1 on the validity of the results. All tests were done in the laboratory of Pasteur Institute, Madagascar using reagents that were provided by the Institute of Tropical Medicine (ITM), including the two monoclonal antibodies that are produced at ITM. Before the start of the study, laboratory technicians of Pasteur Institute received a two-weeks training on the test in Madagascar by an ITM technician. Standard operating procedures were provided to the Pasteur lab and the results of each plate were sent to ITM for quality control. The procedure involves the use on each ELISA plate of a substrate control, a conjugate control, two positive control serum samples and a set of 8 negative control serum samples; the optical densities of the latter being used for the calculation of the cut-off as described in the manuscript. Both the positive and negative controls were provided by ITM. The two positive controls are diluted serum samples from highly infected individuals. They are used as a quality check on the plates. The 8 negative controls originate from non-infected individuals of Belgian origin, and are the same samples than those used at ITM and consequently served as an extra quality check of each plate. In contrast to antibody tests that may show a background as a result of local conditions such as exposure to other pathogens, no such a background has been observed in the Ag-ELISA following the routine TCA pre- treatment of the serum samples before application in the ELISA.We therefore believe that our laboratory results are robust enough to make the conclusions as described in the paper.

We have made changes accordingly in the Materials and Methods section.

Reviewer #2: The manuscript needs a MAJOR revision.

Major comments:

1. Abstract: Should make clear that the samples were from 13 years ago (assuming that is the case), so the conclusion might not be valid, unless they speculate that risk factors haven’t changed.

The sera were collected in 2007. They were stored at 2-8°C for a maximum of 48h then sent to Pasteur Institute of Madagascar where they were kept frozen at -20°C.The T. solium Antigen analysis were performed in 2008. According to Word Bank data (accessed 12/21/2020): GDP per capita (current USD) decreased from 536 to 523 USD (https://data.worldbank.org/indicator/NY.GDP.PCAP.CD?locations=MG). Assuming that the risk factors are associated with poverty and haven’t significally changed, the obtained data may be extrapolated to the current years.

We have made changes accordingly in the conclusions of the discussion section.

2. Study design is not clear, it is only a geographical description.

This is a cross-sectional studyin schoolchildren from seven sites in the four different biotopes of Madagascar.This research was not an ecological study but a cross-sectional study. Therefore, this study was not based on aggregated data and individual data were directly analyzed. The title has been changed in order to be more specific “school children, antigen prevalence study”.

3. Timelines are not clear through the manuscript. 

When were the samples collected? Between February and March 2007 When were the tests performed? Tests have been performed in 2008.

4. Analytical procedures:

4a. Which methods were used to develop the ELISA? The Ag-ELISA hasn’t been developed but implemented at the IPM. It was developed at the ITM, Antwerp. The reagents were provided by the ITM(see response to comments of reviewer#1).

4b. Where were the reagents produced? at the ITM, Antwerp(see response to comments of reviewer#1).

4c. Source of positive controls? ITM, Antwerp (see response to comments of reviewer#1).

4d. Source of negative sera? Are they from the same population? ITM, Antwerp (see response to comments of reviewer#1).

4e. Any positives confirmed as positives? Since no gold standard is available for the diagnosis of Neurocysticercosis, positivity is assessed by clinical, imaging and pathological or biological data (Del Brutto criterions).

5. Statistical analyses:

5a. The analysis by age is not clear. Which groups were analysed? What are the results by group?

This information is not clearly enough in the paper. The age was collected as a continuous variable, and was then treated as a dichotomous variable. We used the population mean as a cut-off and chose the "under 8" category as the reference variable during the analysis. This analysis was also conducted according to the age status. The results are presented in a supplementary table.

Characteristic Categories Total Ag + Ag - 

 n % n % n % P-Value

Sex Male 887 50.83 250 51.76 637 50.48 0.63

 Female 858 49.17 233 48.24 625 49.52 

Age (years) < 8 799 45.79 191 39.54 608 48.18 <0.001

 ≥ 8 946 54.21 292 60.46 654 51.82 

5b. Why try to relate prevalence to altitude and rainfall, and not to the well-known and necessary risk factors such as presence of backyard pigs and open defecation?

As mentioned in the Materials and Methods section the blood samples were collected in the framework of a malaria study. Therefore, factors recorded are not specific to cysticercosis. Moreover, it is well known that open defecation and free roaming of pigs are factors facilitating the transmission of T. solium. In this approach, the objective was to explore the potential influence of environmental characteristics ,altitude, rainfall, bioclimatic patterns. This point is rarely explored. 

6. Results: Results by age are not clear.

Please, refer to 5a. answer.

7. Conclusion:

7a. If the serum is indeed from 2007, 13 years later, many epileptic cases would have been detected. Is this the case? What is the rate of epilepsy? Before the study, all the sera were anonymized, it was not possible to carry out a follow-up. The protocol has been validated by the national ethics committee of Madagascar. The aim being to help the government on the basis of grouped data to take appropriate action at the local, regional or national level. No data regarding this study are therefore available for assessing the epilepsy rate. Furthermore, in some cases blood positivity may not be related to a neurological localization.

8. Ethics: need to justify why the children were not followed-up as NCC is potentially very dangerous (I assume it is because of timings, but as they are not specified, it is not clear). Before the study, all the sera were anonymized, it was not possible to carry out a follow-up. The protocol has been validated by the national ethics committee of Madagascar. The aim being to help the government on the basis of grouped data to take appropriate action at the local, regional or national level.

Minor comments:

9. Line 54: Use original reference for 30% seizure disorders (Ndimubanzi et al, 2010). We thank you for your input, the modification has been made.

10. Line 56: Andean area of South America is not correct (Argentina is not endemic for example). Better change to areas of South America. We thank you for your input, the modification has been made.

11. Figure 1: There is no legend to interpret the colours. The dark blue is difficult to read. What does it mean 237 in the title? We thank you for your input, 237 mention was an error, the modifications have been made.

12. Table 1: Only last row is Ag prevalence. Title is not appropriate. Delete % sign from first results column. We thank you for your input, the modification has been made and the title has been modified accordingly.

13. Editing – in several places spaces before brackets are missing. We thank you for your input, the modification has been made.

Reviewer #3: This is a very interesting manuscript showing that cysticercosis is infecting children at a very high prevalence in urban areas of Madagascar. The authors should however consider the following to improve it

1) the authors should follow the STROBE guidelines for reporting observational studies in epidemiology.The STROBE guideline for cross-sectional studies has been followed and as for this new revised manuscript, modifications have been done accordingly

2) the sampling frame and study sites are not clearly described; there is a mismatch between the text and Fig1, and Fig1 title is unclear. How many sites in the highlands? Why more than 2 sites in the other areas? In each biotope two sites (=town) were selected except for highlands (=1 town) for logistical reasons. We took your input into account: the figure has been improved, all seven study sites have been included. The title has been clarified.

3) given the high prevalence of infection, a log-binomial model would be more appropriate here and make interpretation of ratios more straightforward. As you advised us, we carried out the analysis from a log-binomial model. The results of that model were similar to those obtained with the multilevel logistic regression model.

It is also unclear if the biotype was included as a level in the hierarchical model or as a variable (only 2 levels are mentioned in the description). Each level used in the cluster sampling should be included in the multi-level model. it is unclear if this was done or not. Thank you for that comment. As described in Table II, the variable "Biotype" consists of four modalities. The "Highlands" biotype is the reference modality for the hierarchical model. This variable was introduced as a fixed effect and did not define a hierarchical level (in part because of the very low number of different level units (one of the conditions to perform such models is a minimum of units of each level). Only two levels were available and were introduced in the multilevel model ; level one : child and level two : city. The school level was not entered in the database. The following text was inserted in the manuscript, line 137, page 14 : « Sex, age, biotope, altitude, temperature and rainfall were introducted in the models as fixed effects. »

4) More details on sampling of households and children within each town (is town=site?) are needed. At this point, it is unclear if towns were randomly selected and how children within town were selected. This could introduce selection bias. A two-level cluster random sampling (school and classroom), including schoolchildren from either public or private school was used to select the study population. Depending on the site (=town),samples were collected from one (in Tsiroanomandidy and Miandrivazo), two (Andapa, Ejeda and Moramanga) or three schools (Farafangana, Ihosy and Maevatanana). Clarifications have been added to the text line 87-100).

5) It seems very odd for a child aged 3 years old to be considered as school aged. Do 3-year olds go to school in Madagascar? Yes, this data has been confirmed.

6) Denominators and 95%CI should be reported in Table 1 for ease of reading and interpretation. Thank you for this remark. Rainfall, altitude and temperature are contextual variables, i.e. the same value is affecting all the subjects of the same geographical site, so no variability is considered inside each site. The 95% confidence interval around the proportion of Ag+ and the number of missing data was added per site in Table2.

7) Why was the age dichotomized and was effect modification evaluated? As major epidemiological factors, age and sex were introduced in the analysis. These classical variables were retained to ensure the absence of confusion bias. This could indeed be feared for age (significantly linked to seropositive status).The following text was inserted in part of the results section : « As major epidemiological factors, age and sex were introduced in the analysis. »

The used of dichotomized age allowed us to test for different effects, such as interaction and to more easily cope with a potential problem of linearity assumption.

The following text was inserted in the statistical part of the material and methods section : « Interaction between variables in models were tested. None of them were significant or improved the models. » (143-144)

Were there any other confounders assessed? SES for one could be an important confounder of the impact of even the environmental factors. General sanitation even at the site level is also important to adjust for, it may be confounding the association with environmental factors. What about ethnicity? 

A supplementary table showing the distribution of ethnicities is presented here. The dispersion of the population was very important, unbalanced and concerned 20 ethnic groups. We considered it relevant not to include it in our model. The following text was inserted in the results part : « Twenty ethnic groups are represented in the study population. Most of the children where of Merina or Betsileo ethnic group.” » (152-154).

Ethnies (N= 1744) Number Pourcentage

Merina 507 29,07%

Betsileo 417 23,91%

Betsimisaraka 40 2,29%

Sakalava 45 2,58%

Antaisaka 137 7,86%

Antandroy 76 4,36%

Mahafaly 76 4,36%

Vezo 10 0,57%

Bara 49 2,81%

Antakarana 5 0,29%

Antemoro 103 5,91%

Antaifasy 79 4,53%

Masikoro 0 0,00%

Antambohoaka 1 0,06%

Tsimihety 95 5,45%

Tanala 9 0,52%

Bezanozano 2 0,11%

Sihanaka 5 0,29%

Antanosy 11 0,63%

Zafimaniry 4 0,23%

Foreigners 73 4,09%

There needs to be a Table of univariate analyses. Following the STROBE guidelines would help in this regard.

Thank you for that remark. We have added a complementary table describing the explained variables according to sex and age status. Altitude, rainfall, temperature and biotope are contextual variables. Univariable analyses are presented in Table II.

8) a more solid discussion of the impact of the environmental factors is needed with strenghts and limitations

Few data are available on the impact of the environmental factors, such as rain, altitude and temperature, toward parasites survival in the environment. However as quoted, sociological and behavioral practices added to the sanitation amenities may also be related to a specific environment and may play an even bigger role in the process of breaking or enhancing the parasitic cycle.

9) There needs to a paragraph on limitations

This section has been emphasized in the discussion section.

Specific comments

Line 60-61: please provide a range of prevalence found with Ag ELISA in SSA (like what is done for Ab). There is also a relatively recent systematic review of prev of HCC in SSA which could be used here (Coral Almeida 2015),Thank you for your input, it has been taken into account.

Could there be more than 1 child per household? If so, was this considered in the analyses?

Yes, most households do have more than 1 child but this data was not available and couldn’t be handled in our model.

Study design: the DATES of the survey must be reported. The dates are Between February and April 2007, they have been reported.

Table 2 and related analyses: re-run the analyses with the Equatorial region as a reference for easier interpretation.

As requested, Table II has been adapted and presents the equatorial region as a reference. It is now Table III in the new version.

Lines 175-176: this is not quite correct. The EITB performs a lot better in this situation, though not perfect. Also, the AgELISA also performs poorly when there is only one NCC cyst, so this argument is incorrect the way it is stated here.Thank you for these relevant remarks. Some modifications have been applied to the text (189-190).

Lines 206-209: the big difference here is that most other studies were conducted in adults where the effect gender is often modified by age and is an indirect measure of behavior. This should be noted here.Thank you for your input, it has been taken into consideration and the conclusion has been modified accordingly (208-210).

To the Editor: 

If figure 1 is original, then there is no copyright issue. Can you please provide details within your Response to Reviewers document to confirm that the figure used is original.

Madagascar's layer was drawn using geodata from World of Maps ( www.worldofmaps.net), and mapping operations were performed using QGIS 2.18 software. Source: QGIS Development Team. QGIS Geographic Information System. Open Source Geospatial Foundation.; 2009. Therefore, this map is not copyrighted.

---

## [Decision Letter · Decision Letter 1]

13 Jul 2021

PONE-D-20-27048R1

Hyperendemicity of Cysticercosis in Madagascar: Novel insights from a school children population-based antigen prevalence study

PLOS ONE

Dear Dr. Carod,

Thank you for submitting your manuscript to PLOS ONE. After careful consideration, we feel that it has merit but does not fully meet PLOS ONE’s publication criteria as it currently stands. Therefore, we invite you to submit a revised version of the manuscript that addresses the points raised during the review process.

I agree with reviewer # 5  the authors need to discuss the limitations of the study and critically discuss the obtained data. The main limitation is the lack of clinical data and risk factors associated with cysticercosis. This concern was raised in the first peer review and needs to be properly discussed as suggested in the second revision. The authors should also address the possibility of the transient Ts antigen detection and discuss it accordingly.

We look forward to receiving your revised manuscript.

Kind regards,

Fela Mendlovic, MSc, PhD

Academic Editor

PLOS ONE

Reviewers' comments:

Reviewer's Responses to Questions

**Comments to the Author**

1. If the authors have adequately addressed your comments raised in a previous round of review and you feel that this manuscript is now acceptable for publication, you may indicate that here to bypass the “Comments to the Author” section, enter your conflict of interest statement in the “Confidential to Editor” section, and submit your "Accept" recommendation.

Reviewer #4: All comments have been addressed

Reviewer #5: (No Response)

2. Is the manuscript technically sound, and do the data support the conclusions?

Reviewer #4: Yes

Reviewer #5: Partly

3. Has the statistical analysis been performed appropriately and rigorously? 

Reviewer #4: Yes

Reviewer #5: Yes

4. Have the authors made all data underlying the findings in their manuscript fully available?

Reviewer #4: Yes

Reviewer #5: Yes

5. Is the manuscript presented in an intelligible fashion and written in standard English?

Reviewer #4: Yes

Reviewer #5: Yes

6. Review Comments to the Author

Reviewer #4: Dear Author: Your updated the manuscript with preview reviewers suggestions.

Is a great research, congratulations.

Reviewer #5: The paper reports the results of a study carried out in Madagascar to evaluate the prevalence of Taenia solium circulating antigen (Ts-antigen) in school age-children of seven major cities. The presence of Ts-antigen is considered indicative of infection with viable cysticerci of Taenia solium, although it may be a transient phenomenon with possible sero-reversion, hence not necessary established infection. The prevalence of Ts-Ag was 27%, with percentage up to 37% in some children groups. The observed Ts-Ag prevalence rate exceeds all antigen prevalence rates described worldwide, after one village in DRC where a seroprevalence of 21% was recorded.

The obtained results are of some interest but should be presented evidencing the limitations of the study and critically reviewing obtained data.

The study was conducted on serum samples collected in a malaria survey in 2007. Information on risk factors for cysticercosis (e.g presence of backyard pigs and open defecation) was not collected. No clinical data related to cysticercosis (epilepsia) are available neither for tested subjects nor local population. No data are available about teniosis. The authors explored the potential influence of environmental characteristics, altitude, rainfall and bioclimatic patterns.

The first question/doubt could be about the performance of the test, mainly its specificity, but also about the correct procedures of an in-house test. The authors affirm that tests were done by well-trained laboratory technicians of Pasteur Institute (Madagascar) by using reagents provided by the Institute of Tropical Medicine of Antwerp, where the test was developed and validated. Negative and positive controls were included and the procedures were correctly followed. The results of each plate were sent to ITM for quality control. Based on this, results seem to be robust.

The main limitations of the study are the lack of information about risk factors for cysticercosis and clinical data (mainly epilepsia) at the moment of enrollment (2007) and in subsequent years, as well as data on teniosis. It would be interesting to address this gap through other research activities in the studied areas. If data were already available, although preliminary, they could be mentioned in the paper.

Moreover, it is known that the presence of cysticercal circulating antigen may be short-live and sero-reversion is a common phenomenon. The authors mention this point in the final part of the discussion citing the paper of Mwape et al (ref 37). I suggest to read carefully the paper of Mwape et al reporting the results of a longitudinal study in Zambia with blood sampled three times, with a 6-month interval and analyzed for the presence of cysticercal circulating antigens and specific antibodies. Mwape et al paper should inspire the authors to present their results more critically and to revise the discussion.

7. PLOS authors have the option to publish the peer review history of their article (what does this mean?). If published, this will include your full peer review and any attached files.

Reviewer #4: **Yes: **LMMV

Reviewer #5: No

---

## [Author Response · Author response to Decision Letter 1]

22 Aug 2021

Answers to the Editor and to Reviewer(s)

Editor: I agree with reviewer # 5 the authors need to discuss the limitations of the study and critically discuss the obtained data. The main limitation is the lack of clinical data and risk factors associated with cysticercosis. This concern was raised in the first peer review and needs to be properly discussed as suggested in the second revision. The authors should also address the possibility of the transient Ts antigen detection and discuss it accordingly.

Reviewer #5: The paper reports the results of a study carried out in Madagascar to evaluate the prevalence of Taenia solium circulating antigen (Ts-antigen) in school age-children of seven major cities. The presence of Ts-antigen is considered indicative of infection with viable cysticerci of Taenia solium, although it may be a transient phenomenon with possible sero-reversion, hence not necessary established infection. The prevalence of Ts-Ag was 27%, with percentage up to 37% in some children groups. The observed Ts-Ag prevalence rate exceeds all antigen prevalence rates described worldwide, after one village in DRC where a seroprevalence of 21% was recorded.

The obtained results are of some interest but should be presented evidencing the limitations of the study and critically reviewing obtained data.

The study was conducted on serum samples collected in a malaria survey in 2007. Information on risk factors for cysticercosis (e.g presence of backyard pigs and open defecation) was not collected. No clinical data related to cysticercosis (epilepsia) are available neither for tested subjects nor local population. No data are available about teniosis. The authors explored the potential influence of environmental characteristics, altitude, rainfall and bioclimatic patterns.

The first question/doubt could be about the performance of the test, mainly its specificity, but also about the correct procedures of an in-house test. The authors affirm that tests were done by well-trained laboratory technicians of Pasteur Institute (Madagascar) by using reagents provided by the Institute of Tropical Medicine of Antwerp, where the test was developed and validated. Negative and positive controls were included and the procedures were correctly followed. The results of each plate were sent to ITM for quality control. Based on this, results seem to be robust.

The main limitations of the study are the lack of information about risk factors for cysticercosis and clinical data (mainly epilepsia) at the moment of enrollment (2007) and in subsequent years, as well as data on teniosis. It would be interesting to address this gap through other research activities in the studied areas. If data were already available, although preliminary, they could be mentioned in the paper.

Moreover, it is known that the presence of cysticercal circulating antigen may be short-live and sero-reversion is a common phenomenon. The authors mention this point in the final part of the discussion citing the paper of Mwape et al (ref 37). I suggest to read carefully the paper of Mwape et al reporting the results of a longitudinal study in Zambia with blood sampled three times, with a 6-month interval and analyzed for the presence of cysticercal circulating antigens and specific antibodies. Mwape et al paper should inspire the authors to present their results more critically and to revise the discussion.

We thank you for your input. Although specific risk factors for cysticercosis were not available in this study, other relevant environmental factors were discussed as possible risk factors. The main benefits of this study are that

- we benefited from an extensive amount of multi-centered sera collected in 2007 amongst whole schoolchildren communities

- we used the B158/B60 Ag-ELISA, a diagnostic tool for cysticercosis that has proven to produce valuable results in many studies but that has never been used in Madagascar.

As stated by both Editor and reviewer #5, we acknowledge the lack of both clinical and specific risk factors. These could have enabled us to better understand our results. As suggested, we rewrote our discussion, strengthening on the limitations of our study and trying to link our results with available information on epilepsy in Malagasy children.

We have made changes accordingly in the discussion section, as follows (in red in the text):

Discussion

The seroprevalence of cysticercosis based on antibody detection ranked Madagascar within the highly endemic countries. The lack of sensitivity of Ab-ELISA’s may have underestimated the prevalence rates published in Madagascar. Most NCC cases (the most common clinical form of cysticercosis) found on the island are related to single cyst carriers, which are known to be difficult to detect with both antibody and antigen-based serological methods (26,27).

In contrast to the presence of Ts Ab, indicating exposure to the infection, the Ts Ag rate indicates the presence of active cysticercosis. With an average of 27% positives in children in our study, the Ts Ag prevalence rate exceeds all antigen prevalence rates described worldwide. High Ts Ag rates indicates that active infections with T. solium metacestodes occur within the Malagasy population, affecting the younger age class and that it is widespread on the island even though a visible heterogeneity was observed between the sites. Recent studies have shown that in African countries higher apparent prevalence of active infections occurred than in Asian countries (28). This difference may reflect epidemiological differences, especially a higher exposure; however, variations in host susceptibility and in parasite genotypes may also play a role (29,30). 

Previous evidence based on antibody detection has indicated that all provinces of Madagascar were endemic for cysticercosis with significant differences between the areas. The use of the Ts Ag ELISA showed the same trends with a lower Ts Ag prevalence in the equatorial biotope. This supports the hypothesis that spatial heterogeneity in the distribution of infections may be influenced by environmental conditions, highlighting the interplay between socio-economic, behavioural and environmental factors in Taenia (31). Selection bias may also be taken into account. One major difference of our study is that most other studies were conducted in adults in which the effect of gender is often modified by age and is an indirect measure of behaviour. Biotopes are represented by solely one or two sites with sometimes marked variations in environmental or cultural behaviours. The access to fresh water is enhanced in the equatorial area with better hygiene practices, and less common pig raising and pork consumption than in the highlands (32). These conditions and practices may play a role in limiting the disease’s prevalence. While arid areas should conversely experience a lower survival of parasitic eggs (33), in the present study these areas were associated with high Ts Ag prevalence rates, just below the highlands’ rate. When considering the effect of desiccation on Taenia egg survival (33), this high Ag prevalence was not expected and requires further study.

An effect of gender on prevalence was not observed (20), unlike in studies in Latin America and other African countries. This might be due to cultural differences between populations (28). However, age was reported as a significant risk factor in the bivariate model for the presence of circulating antigens as quoted in different reports (28,34,35). Worldwide, one third of the total epilepsy cases arise in childhood and neurocysticercosis is a major cause, particularly in developing countries (35,36). However, presence of antigen doesn't necessarily signify the presence of a viable, well-established cysticercus infection. It could be indicative for short term partial establishment, and perhaps a ‘transient’ antigen presence should be considered (37). A Zambian study demonstrated the rather short-term presence of circulating antigen in 31 participants. Whether this observation is due to a partial establishment of infection (immature cysticerci), or the establishment of cysticerci, followed by a quick degeneration (self cure?) could not be confirmed. The Ts Ag results in that study were provided as qualitative data. It may be that antigen levels (ratio) can add complementary information. Indeed, Mwape et al (37) observed that individuals who became seronegative were those that displayed low levels of circulating antigen. Therefore, serological results from field studies should be looked at critically. Individuals with positive test results shouldn't be by definition considered as infected with T. solium, but the possibility of a transient antigen response has to be taken into account.

The benefit of treating children positive for Ts antigen is still debated, and the medical interventions should be carried out in neurology Units. However, in Madagascar neurologists are available only in Antananarivo and Mahajunga, and not in the cities under study. Treating a positive case without a proper medical examination may impair and even endanger the child’s life and by destroying viable cysts, it may enhance the brain seizures with a high risk of morbidity and even mortality (9). 

Limitations

This is a cross-sectional study in schoolchildren from seven sites in the four different biotopes of Madagascar. The samples were collected 13 years ago. It may be suspected that risk factors have changed. However, according to Word Bank data (accessed 12/21/2020): GDP per capita (current USD) decreased from 536 to 523 USD. Assuming that the risk factors are associated with poverty and haven’t significantly changed, the obtained data may be extrapolated to the current years. 

The main limitations of our study are the lack of clinical data and risk factors associated with cysticercosis.

- The blood samples were collected in the framework of a malaria study. Therefore, factors recorded were not specific to cysticercosis. Further and updated studies should rely on prospective studies focusing on social, behavioural, sanitary, economic determinants and the Taeniasis/Cysticercosis risk factors that may play a role in the spread of the disease and explain the differences between sites. Implementing specifically cysticercosis driven prospective studies will enable to show the dynamic nature of T. solium infections in the different biotopes and adapt accordingly human and animal health policies.

- Clinical features have to be taken into account and a possible follow-up should be offered to patients featuring a probable/possible diagnosis of NCC (38). The presence of a positive Ag Ts is not sufficient to diagnose NCC since its positivity does not indicate a neurological localization and an important number of infections probably never fully establish, leading to possibly ‘transient’ antigen presence (37). Furthermore, in humans, it is described that cysticerci may stay viable during years according to their localization (3). Many factors, among which, the size of the (re) infection, the immune status of the host, age and sex play a determining role in the (non-) establishment of infection (39). 

Clinical features including epilepsy are critical to define a diagnosis of NCC added to imaging data, laboratory findings and epidemiology patterns (38). Many reports described cysticercosis-associated epilepsy in Madagascar (40): pediatricians of the military hospital of Antananarivo reported that epilepsy was present in over 80% of the cases of NCC. An interesting finding is that the seroprevalence (Ts Ab) found among suspected cases was between 25 and 48% (ELISA+/-EITB) (40). However, those studies were completed between 1991 to 1993 and did not use Ts Ag as the diagnostic tool. Therefore, unpublished data communicated in 2012, displayed a Ts Ag positivity of 73% among patients having clinical and imaging features attributable to NCC (41). Clinical symptoms and biological results are, however, not sufficient for a diagnosis of NCC (38). CT-scan/MRI are essential for this purpose: they are the sole way to assess the number, the neurological localisation and the status (viable/degenerated/calcified) of cysticerci. Unfortunately, these imaging tools are obviously not applicable and available for field studies.

---

## [Decision Letter · Decision Letter 2]

17 Sep 2021

Hyperendemicity of Cysticercosis in Madagascar: Novel insights from a school children population-based antigen prevalence study

PONE-D-20-27048R2

Dear Dr. Carod,

We’re pleased to inform you that your manuscript has been judged scientifically suitable for publication and will be formally accepted for publication once it meets all outstanding technical requirements.

Within one week, you’ll receive an e-mail detailing the required amendments. When these have been addressed, you’ll receive a formal acceptance letter and your manuscript will be scheduled for publication. Please check the word spacing and some of the verb tenses that are incorrect.

Kind regards,

Fela Mendlovic, MSc, PhD

Academic Editor

PLOS ONE

Additional Editor Comments (optional):

Reviewers' comments:

Reviewer's Responses to Questions

**Comments to the Author**

1. If the authors have adequately addressed your comments raised in a previous round of review and you feel that this manuscript is now acceptable for publication, you may indicate that here to bypass the “Comments to the Author” section, enter your conflict of interest statement in the “Confidential to Editor” section, and submit your "Accept" recommendation.

Reviewer #5: All comments have been addressed

2. Is the manuscript technically sound, and do the data support the conclusions?

Reviewer #5: Yes

3. Has the statistical analysis been performed appropriately and rigorously? 

Reviewer #5: Yes

4. Have the authors made all data underlying the findings in their manuscript fully available?

Reviewer #5: Yes

5. Is the manuscript presented in an intelligible fashion and written in standard English?

Reviewer #5: Yes

6. Review Comments to the Author

Reviewer #5: (No Response)

7. PLOS authors have the option to publish the peer review history of their article (what does this mean?). If published, this will include your full peer review and any attached files.

Reviewer #5: No

---

## [Editor Report · Acceptance letter]

23 Sep 2021

PONE-D-20-27048R2 

Hyperendemicity of Cysticercosis in Madagascar: Novel insights from school children population-based antigen prevalence study 

Dear Dr. Carod:

I'm pleased to inform you that your manuscript has been deemed suitable for publication in PLOS ONE. Congratulations! Your manuscript is now with our production department. 

Kind regards, 

on behalf of

Dr. Fela Mendlovic 

Academic Editor

PLOS ONE